# Replication Study of "Fairness and Bias in Online Selection"

1    **Reproducibility Summary**

2    **Scope of Reproducibility**

3    In this paper, we work on reproducing the results obtained in the 'Fairness and Bias in Online Selection' paper (Correa,
4    Cristi, et al., 2021). The goal of the reproduction study is to validate the 4 main claims made in Correa, Cristi, et al.
5    (2021). The claims made are: (1) for the multi-color secretary problem, an optimal online algorithm is fair, (2) for
6    the multi-color secretary problem, an optimal offline algorithm is unfair, (3) for the multi-color prophet problem, an
7    optimal online algorithm is fair (4) for the multi-color prophet problem, an optimal online algorithm is less efficient
8    relative to the offline algorithm.

9    To test if the results of the secretary algorithm generalize to other data sets, the proposed algorithms and baselines are
10   applied to the UFRGS Entrance Exam and GPA data set (Castro da Silva, 2019).

11   **Methodology**

12   The paper that has been reproduced includes a link to a repository containing $C++$ files for the algorithms that were
13   implemented. For our experiments, we reimplemented the code in *Python*. Our goal was to reproduce the code in an
14   efficient manner without altering the core logic. Using the Python code all the experiments in the paper have been
15   replicated including some additional experiments to verify the claims made in Correa, Cristi, et al. (2021).

16   **Results**

17   The reproduced results support all claims made in Correa, Cristi, et al. (2021). However, in the case of the unfair
18   secretary algorithm (SA), some irregular results arise in the experiments due to randomness. This irregularity is also
19   existent in the original code.

20   **What was easy**

21   The concepts behind the algorithms were straightforward. The existing code base provided a solid reference point to
22   verify the results of the original paper by compiling and running the provided code.

23   **What was difficult**

24   Implementing the prophet algorithm, in comparison to the secretary algorithm, was complex. $C++$ is a more efficient
25   compiler (time complexity, etc.) compared to Python. For the reproduction of the algorithms, this needed to be taken
26   into account. While it might be possible to execute transliterated code on a powerful machine, with the available
27   resources the code would have taken over 96 hours to run. In order to tackle this problem, some of the data structures
28   needed to be converted to *NumPy* arrays to decrease computation time.

Submitted to ML Reproducibility Challenge 2022. Do not distribute.

# 1  Introduction

As more machine learning algorithms are used in decision-making circumstances, it is important to ensure that social norms are not violated. The social norm that serves as the pivot of this research is fairness. Specifically 'fairness' in the use of selection models. The importance of fairness is to avoid undesirable biases. Selection models are models that input a finite amount of agents and attempt to pick the best possible candidate (agent). The goal is to design algorithms that can fairly judge between agents regardless of any unfair bias.

In some real-life implementations of selection models, there is no clear overview of all agents. For example, in the online selection problem, the agents enter the algorithm sequentially. For every agent, a decision has to be made whether this is the best possible agent. The complexity of this task is not being able to have any knowledge on agents that might come in the future. As soon as the decision is made that an agent is the best fit, the algorithm should stop as that agent is the optimal candidate (according to the model). Multiple attempts have been made to create the most accurate algorithm for these online selection models.

For this research, we reproduce the 'Fairness and Bias in Online Selection' paper (Correa, Cristi, et al., 2021). In this paper, the authors focus on 2 main problems: the secretary problem and the prophet problem. The secretary problem is a scenario for the sequential selection problem where an attempt is made to select the candidate with the highest value without knowing the value of the candidates to come. An immediate decision has to be made on the candidate, the candidate either gets picked or gets passed on. For the prophet algorithm the same assumptions are made as for the secretary algorithm, but we know the distributions the candidate values are drawn from. The probability of the candidate is based on these distributions. In the case of both problems, the goal is to stop at the best possible candidate based on the assigned probabilities.

In order to include a form of fairness in these models, a concrete definition needs to be given to fairness in online selection models. Based on the Correa, Cristi, et al. (2021) paper, fairness is defined as an unbiased evaluation of agents in a selection model. A selection algorithm is fair if it selects the best candidate, closely following the original probability of the best candidate existing in that group. Along with fairness, efficiency has also been used as an evaluation metric in the original paper. Efficiency is a measure of how accurately the online algorithm picks the actual best candidate.

By creating a 'fair' version for these problems, the authors claim to have created a fair use of sequential single item selection models. Through categorization of the agents by color, a distinction between the agents can be made. However, the qualities these agents possess might be different enough that they could be considered incomparable. So implementing a multi-color version of the sequential selection models and picking the best possible candidate, taking color into account, an 'unfair' comparison is avoided.

# 2  Scope of reproducibility

In this reproduction study, we focus on the authors' claims that the use of a multi-color version of the secretary and prophet problem would make the use of these algorithms fair. The authors of the paper implement these algorithms on synthetic data sets and real-world data sets.

For our study, we put an effort into reproducing the results given by the paper. The goal of this reproduction is to either validate or deny the claims made in the paper. This effort has been fulfilled by re-implementing the code publicly available for the algorithm. This re-implementation is done in *Python* in comparison to the *C++* code provided by the authors. Most of the code has been written using *NumPy* to try and achieve about the same efficiency as the *C++* code. However, the setup for the experiments corresponds to that of the authors.

To show that the claims generalize well over differently distributed data sets, we run the proposed algorithms and baselines on the UFRGS Entrance Exam and GPA data set (Castro da Silva, 2019).

The claims made in the Correa, Cristi, et al. (2021) paper are:

- Claim 1: For the multi-color secretary problem, an optimal online algorithm is fair.

- Claim 2: For the multi-color secretary problem, an optimal offline algorithm is unfair.

- Claim 3: For the multi-color prophet problem, an optimal online algorithm is fair.

75 • Claim 4: For the multi-color prophet problem, an optimal online algorithm is less efficient relative to the
76 offline algorithm.

77 To test these claims we use the algorithms mentioned above on 4 types of data sets. These data sets are further discussed
78 in section 3.3.

## 3 Methodology

80 In this section, our approach to the re-implementation of the experiments will be discussed and an additional experiment
81 will be proposed.

### 3.1 Code

83 The code accompanying the paper is provided in $C++$. As required for this study, we reproduced the work in Python,
84 and subsequently made use of the inherent Pythonic efficiencies. The provided code allowed for a smooth initial
85 reproduction. However, many optimisations were required to decrease computation duration.

### 3.2 Model descriptions

87 In the original paper, two types of single item selection models are considered: the secretary algorithm and the prophet
88 algorithm. Candidates are partitioned into different groups which the authors refer to as *colors*. Every candidate has
89 a numerical value that indicates the capabilities of that candidate. The authors refer to these indicators as *values*.
90 Candidates arrive sequentially, and upon arrival, the algorithms decide whether the candidate is the best candidate
91 overall. The best candidate is defined as the candidate with the highest value of the sequence of candidates. For clarity,
92 the main parts of the Methodology and Results sections are divided per model.

#### 3.2.1 Secretary Algorithm

94 For the secretary algorithm, it is assumed that candidates arrive in uniformly random order. To verify the claims
95 made by the author, we compare the optimal online algorithm as proposed by Correa, Cristi, et al. (2021) to two
96 baselines. Additionally, the algorithm and its baselines are applied on different data sets, either synthetically generated
97 or composed from real-word data sets. The optimal online algorithm proposed by the authors (Fair secretary algorithm)
98 is denoted formally as:

---
**Algorithm 1** GROUPTHRESHOLDS($\mathbf{t}$)

---
**Input:** $\mathbf{t} \in [0,1]^k$, a threshold in time for each group
**Output:** $i \in [n]$, index of chosen candidate

```
/* assuming arrival times τ₁ < ... < τₙ  */
```
**for** $i \leftarrow 1$ **to** $n$ **do**
   **if** $\tau_i > t_{c(i)}$ **then**
      **if** $i \succ \max\{i' \mid \tau_{i'} \leq \tau_i, c(i') = c(i)\}$ **then**
         **return** $i$
      **end**
   **end**
**end**

---

99 where the input $\mathbf{t} = (t_1, ..., t_k)$ is a vector of thresholds, one for each color $j \in [k]$. The algorithm first checks if the
100 candidate $i$ arrived after the threshold of its color $t_{c(i)}$. If this condition is met, it accepts the candidate if its value
101 exceeds the value of all previous candidates of color $t_{c(i)}$, indicating that it is the best candidate for that color.

102 After having chosen the best candidate of each color, we are interested in selecting the best overall candidate. We
103 denote the probabilities with which the best candidate of group $j$ is the best among all colors by $p_j$, which results in the
104 vector $\mathbf{p} = (p_1, ..., p_k)$ covering all colors. We use this in our experiments to verify the claims of the author using equal,
105 and unequal values for $\mathbf{p}$ among colors.

### 3.2.2 Prophet Algorithm

For the prophet algorithm, the same assumptions are made as for the secretary algorithm, but we know the distributions $F_i$ the candidate values are drawn from. In the paper, the authors propose two optimal online algorithms specified in figure 1, where $q_1, \cdots, q_n$ denote the marginal probabilities that the optimal fair offline algorithm picks the candidates $i = 1, \cdots, n$. Figure 1a shows the general Fair prophet algorithm (Fair prophet algorithm). This algorithm does not make any assumptions about the underlying probability distribution, it can be different for every candidate. Figure 1b shows the Fair independent and identically distributed prophet algorithm (Fair IID prophet algorithm). This algorithm assumes that the values of the candidates are drawn from the same distribution.

---

**Algorithm 2** FAIR GENERAL PROPHET

**Input:** Distributions $F_1, \cdots, F_n$, and $q_1, \cdots, q_n$
**Output:** $i \in [n]$, index of chosen candidate

$s \leftarrow 0$
  **for** $i \leftarrow 1$ **to** $n$ **do**
    **if** $v_i \geq F_i^{-1}(1 - \frac{q_i/2}{1-s/2})$ **then**
      | **return** $i$
    **end**
    $s \leftarrow s + q_i$
**end**

(a) Fair prophet algorithm

**Algorithm 3** FAIR IID PROPHET

**Input:** Distributions $F$
**Output:** $i \in [n]$, index of chosen candidate

**for** $i \leftarrow 1$ **to** $n$ **do**
  **if** $v_i \geq F^{-1}(1 - \frac{2/3n}{1-2(i-1)/3n})$ **then**
    | **return** $i$
  **end**
**end**

(b) Fair IID prophet algorithm

Figure 1: Fair prophet algorithms proposed by the authors.

### 3.3 Data sets

The experiments involving the SA algorithm are conducted on two synthetic data sets and two real-world data sets. The data sets and their properties are summarised below:

1. **Synthetic data set, equal p values** contains four different colors with 10, 100, 1000, and 10000 occurrences. The value of each element is chosen independently and uniformly at random from $[0, 1]$.

2. **Synthetic data set, general p values** contains a similar setup as 1, but with $p = (0.3, 0.25, 0.25, 0.2)$.

3. **Feedback maximization (Bank)** contains records of direct marketing campaigns (phone calls) by a Portuguese banking institution (Moro et al., 2014). The clients are split into 5 colors by age: under 30, 31-40, 41-50, 51-60, and over 61 years old. The value of every client is the duration of the phone call. Moreover, an equal $p$ of 0.2 was used for all colors.

4. **Influence maximization (Pokec)** contains records of the influence of users of the Pokec social network (Takac & Zábovský, 2012). We pre-process the data by dividing the users into 5 different colors according to their body mass index (BMI): under weighted (BMI < 18.5), normal (18.5 <= BMI < 25), over weighted (25.0 <= BMI < 30.0), obese type 1 (30.0 <= BMI < 35), and obese type 2 (BMI >= 35.0). The value is computed as the number of the followers for each user. Again, an equal $p$ of 0.2 was used for all colors.

### 3.4 Experimental setup

In this subsection, the experimental evaluation performed by the authors is discussed. As before, a distinction between the two problems is made for clarity. Additionally, an extra experiment will be considered where the secretary algorithm will be evaluated on another real-world data set.

**Secretary experiments**

The authors propose two different baselines to compare the Fair secretary algorithm to. Firstly, the classic secretary algorithm (SA), which does not take the colors of the candidates into account. Secondly, the single-color secretary algorithm (SCSA). This algorithm picks a color proportional to the *p* values and then runs the classic secretary algorithm on the candidates of only that color. To evaluate the claims by the authors, the three mentioned algorithms are evaluated against the four data sets discussed earlier.

The parameters of these experiments consist of the size of the data sets and the number of repetitions. For the experiments on the Synthetic data sets (equal *p* / general *p*) and the Bank data set, all available candidates were used in 20.000 repetitions. In the original paper, the authors used all $\pm$ 650.000 candidates of the Pokec data set in 1000.000 repetitions. In our experiment, we had to limit these parameters due to time constraints. We only considered the first 40.000 candidates and used 40.000 repetitions.

**Prophet experiments**

For the prophet experiments, the Fair prophet algorithm and Fair IID prophet algorithm are evaluated against three baselines: the SC algorithm (Samuel-Cahn, 1984), EHKS algorithm (Marx, 2021), CFHOV algorithm (Correa, Foncea, et al., 2021) and DP algorithm (Brown, 1972). The specific works of these algorithms are described in further detail in the paper (Correa, Cristi, et al., 2021) section 4.2.

For the experiments, two settings are implemented. In the first setting, 50 samples are taken from a uniform distribution in a range of [0, 1]. These samples function as the input stream. In the second setting, 1000 samples are taken from a binomial distribution with 1000 trials and a probability of a successful single trial $p = 0.5$. In order to compare this method with the already existing algorithms, we assume each candidate to be group of its own. For every algorithm, we repeat the experiment 50.000 times.

**Extending to other data set (UFRGS) experiments**

This subsection describes an experimental extension on the work of Correa, Cristi, et al. (2021). In our work, we have concluded that the secretary results claimed in the paper are reproducible. It is shown in section 4 that the Fair algorithm significantly outperforms the SCSA baseline. However, all real-world data sets used to prove this claim contain the same distribution of values for every color. The distributions for the Bank and Pokec data sets are shown in Figures 2a 2b respectively.

Our extension investigates the effect of applying the Fair algorithm to an unequally distributed real-word data set, such as the UFRGS Entrance Exam and GPA Data (UFRGS) data set. This work will show whether the claims made by the authors generalize to these types of data sets. The UFRGS contains entrance exam scores of students applying to a university in Brazil (Federal University of Rio Grande do Sul), along with the students' GPAs during the first three semesters at university. The data set also includes the gender of every student (male or female). The distribution of the data set is shown in Figure 2c. This experiment is a duplication of the original secretary experiments but with the UFRGS data set as input. The gender of the students is used as color, their GPA score as values. The experiment is repeated 20.000 times.

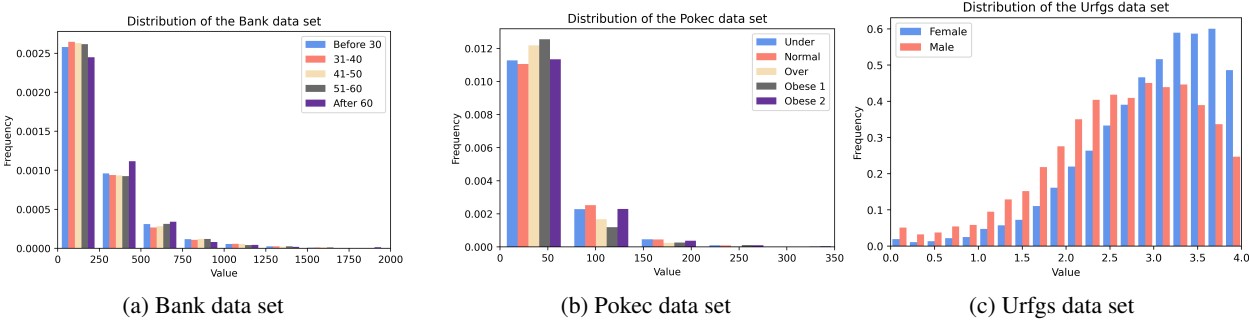

(a) Bank data set        (b) Pokec data set        (c) Urfgs data set

Figure 2: Value distributions of the different color groups in the real-world secretary algorithm data sets.

## 4 Results

The following paragraphs will present the results for the experiments discussed in section 3.4: (1) the secretary experiments, (2) the prophet experiments, (3) our extended work.

**Secretary results**

The plots in Figure 3 show our reproduction work regarding the original paper on the secretary problem over the four different data sets. We find that all results are in line with the work of Correa, Cristi, et al. (2021). Due to the nature of construction of the fair algorithm proposed by the authors, and the SCSA, we find that it picks elements from each color proportional to the vector **p**. From this, it can be concluded that the authors' Claims 1 and 2 are valid.

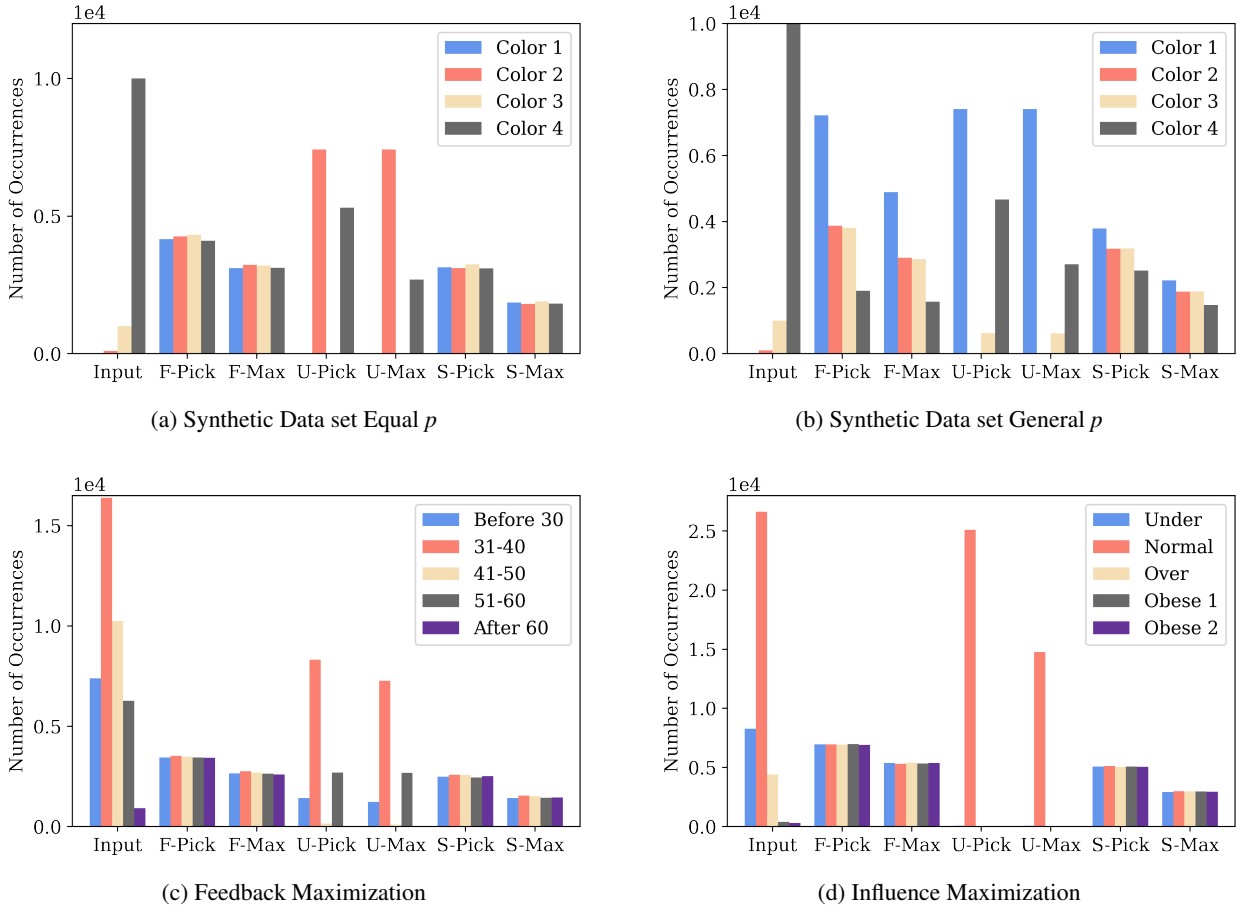

Figure 3: Reproduction work regarding the original paper on the secretary problem. Comparing the Fair secretary algorithm to the aforementioned baselines SA, SCSA over four different data sets: (a) synthetic data set, equal *p* values, (b) synthetic data set, general *p* values, (c) feedback maximization data set (Bank), and (d) influence maximization data set (Pokec). Input denotes the number of elements from each color in the input, F-Pick and F-Max are the number of elements picked by the fair secretary algorithm and the number of them that are the maximum among the elements of that color. Similarly, U-Pick (S-Pick) and U-Max (S-Max) are the number of elements picked by SA and SCSA and the number of them that are the maximum among the elements of that color

The authors claim that the quality of the solution of their algorithm is significantly higher than the SCSA. Table 1 shows our replication of this comparison. We find that our implementation reproduces the authors' claim that their method is superior to the SCSA. Small discrepancies in the results are found, this is due to the random nature of the algorithm. However, as mentioned earlier, after scrutinizing the distributions of the used data sets, we found that all the used data sets have similar distributions in the input. Therefore, we proceed by agreeing with the claims of the author given this restriction.

| Data set | Claimed Pick | Reproduction Pick | Claimed Max | Reproduction Max |
|---|---|---|---|---|
| Synthetic (Equal $p$) | 1.305 (+30.5%) | 1.326 (+32.6%) | 1.721 (+73.1%) | 1.685 (+68.5%) |
| Synthetic (General $p$) | 1.309 (+30.9%) | 1.334 (+33.4%) | 1.630 (+63.0%) | 1.666 (+66.6%) |
| Bank | 1.347 (+34.7%) | 1.377 (+37.7%) | 1.760 (+76.0%) | 1.812 (+81.2%) |
| Pokec | 1.373 (+37.3%) | 1.368 (+36.8%) | 1.756 (+75.6%) | 1.810 (+81.0%) |
| UFGRS | - | 1.192 (+19.2%) | - | 1.364 (+36.4%) |

Table 1: Secretary experiment claims by the author compared to reproduced results.

## Prophet results

The patterns of the results in the original paper are reflected in our reproduction as visualized in figured 4. A major difference is that the scale of their y-axis is twice the size of our reproduction. Because the shown plots are a histogram of arrival positions, this could be attributed to a difference in bin size. The authors' report specifies using uniform distributions. Table 2 shows our replication of the average values chosen by each algorithm. While small differences exist, our reproduction mirrors the authors' results upon running their code closely.

| Algorithm | Uniform Distribution | | Binomial Distribution | |
|---|---|---|---|---|
| | Claimed value | Reproduction value | Claimed value | Reproduction value |
| Fair PA | 0.501 | 0.497 | 0.297 | 0.273 |
| Fair IID | 0.661 | 0.654 | 0.389 | 0.364 |
| SC | 0.499 | 0.494 | 0.227 | 0.253 |
| EHKS | 0.631 | 0.625 | 0.362 | 0.339 |
| CFHOV | 0.752 | 0.755 | 0.513 | 0.408 |
| DP | 0.751 | 0.752 | 0.429 | 0.340 |

Table 2: Prophet experiment claims by the author compared to reproduced results.

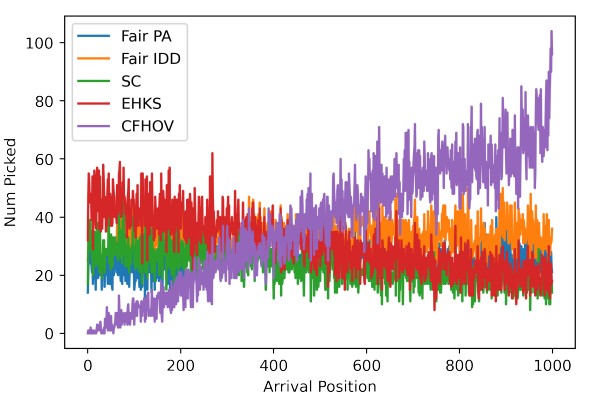

(a) Reproduced Binomial Distribution

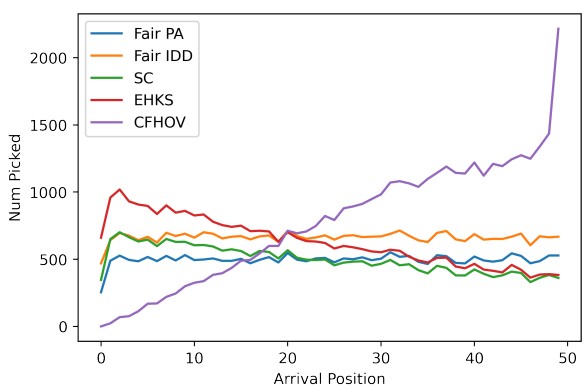

(b) Reproduced Uniform Distribution

Figure 4: Reproduced results for the prophet experiments.

## Extending to other data set (UFRGS) results

Figure 5 shows the results of the experiments proposed in section 3.4. It can be noted that the pattern visible in the earlier secretary results still holds for a new, unequally distributed data set. However, when looking at Table 1, a significant decrease in performance can be detected. The Bank and Pokec data sets scored +37.7% and +36.8% for F-Pick compared to S-Pick. The UFRGS only has an increase of +19.2%. The difference is even more significant when comparing F-Max to S-Max; Pokec and Bank have an increase of +81.2% and +81.0%, UFRGS only has an improvement of +36.4%. We can conclude that the performance increase of the Fair secretary algorithm is not as significant when using an unequally distributed data set, compared to the increase mentioned in the paper.

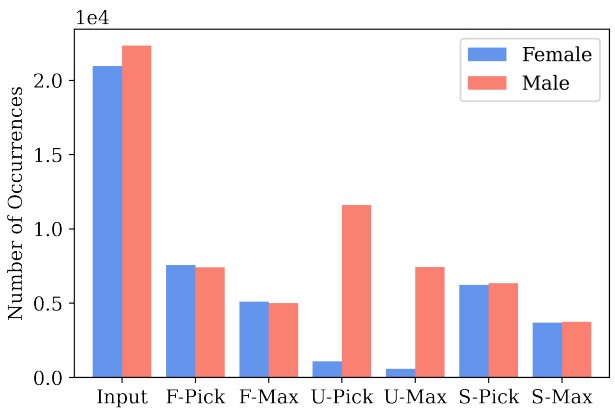

Figure 5: Secretary experiment applied to the UFRGS data set.

## 5 Discussion

In this research, we have tried to reproduce the work of Correa, Cristi, et al. (2021) as closely as possible. However, there are a few inconsistencies in the original code and paper, which caused complications. These points and our solution to them (if required) will be briefly discussed in the following paragraph.

Firstly, as mentioned before, the BMI thresholds for the pre-processing of the Pokec dataset were missing in the authors' work. This poses a problem as slight alterations to these thresholds yield different results. This problem was solved by finding concurring values in other research. Secondly, to limit the computation time of our reproduction, the size of the Pokec data set was limited from approximately 650.000 to 40.000 elements. The number of repetitions for this experiment was also decreased from 1.000.000 to 40.000. We opted for this solution as the distributions in the results did not change from these limits onward. Thirdly, the U-pick/U-max values in the secretary results of the original work are inconsistent due to randomness. It seems that changing the seed value of the random number generator in the *C++* code heavily changes the output of the SA algorithm (U-pick/U-max). The SA results could therefore be cherry-picked as no further explanation was provided by the authors. Lastly, some inconsistencies are present in the paper. From minor typos e.g. using the word *desbribed* instead of *described*, to more serious mistakes, such as claiming that an increase of 1.721 is equal to (+73.1%). A thorough reread of the paper would have resolved this.

### 5.1 Reflection on our replication study

The algorithms used in the original were clear and straightforward. The existing *C++* code of the authors provided a good starting point for the verification of the results.

However, our goal was to further validate these claims and generalize them to a further extent. We did this by reproducing the work of the original paper. Reproducing the work efficiently in another language, in our case *Python*, introduced some difficulties and took longer than expected. An execution of transliterated code resulted in an excessive run time. To tackle this problem, some of the data structures needed to be converted to *NumPy* arrays to decrease computation time. This requires advanced knowledge of *Numpy* and the use of data structures.

### 5.2 Communication with original authors

As certain parameters and split-off values were not clearly defined in either the paper or the original code, we reached out to the authors via mail to ensure a fair assessment of the reproduction. Examples of missing split-off values are the BMI category thresholds for the pre-processing of the Pokec data set. These category values are not fixed in literature and differ depending on age and nationality. At the time of writing this report, we had not yet heard back from the authors. We resolved this by assuming certain values and explanations, which are all documented in our paper.

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
