# OpenReview forum: "Replication Study of "Fairness and Bias in Online Selection""
_ML_Reproducibility_Challenge/2021/Fall — RC2021_

### Official Review · Reviewer_RbQm · 2022-02-18
**Thorough reproduction**

**Rating:** 8
**Confidence:** 4

**Review:**

I'd like to start by thanking the authors for the effort invested in reproducing the results in the paper. I thoroughly appreciate your work.

I found the report to be thorough and informative. The problem tackled and claims in the original paper are described in detail, which substantially helps understanding the experiments involved. The claims checked are accurately described and the report provides sufficient evidence to support the majority of the claims in the paper. The addition of the experiments on the UFRGS is an additional bonus to verifying the claims in the paper. The authors took the time to reimplement the original codebase from C++ to Python and checked the effect of the random seed on the performance of the C++ code, finding it is a substantial source of noise.

Overall I believe the report manages to validate most of the claims in the original paper and the experimental evaluation was conducted thoroughly and fairly.

---

### Official Review · Reviewer_CHag · 2022-02-28
**Review of Replication Study of "Fairness and Bias in Online Selection"**

**Rating:** 7
**Confidence:** 3

**Review:**

This paper discusses the reproducibility of the paper 'Fairness and Bias in Online Selection' (Correa 2021). Overall, the paper is easy to read and is well organized.

Pros
+ The paper does a good job of covering the main topics of reproducibility summary, scope, communication with original authors, discussion on results, and results beyond the paper.
+ The authors extended the experiments to test on another dataset to see if similar findings can be observed.
+ The authors were able to confirm the claims of the original author of the paper even in another language (C++ to Python).

Cons
- Some of the original results and reproduced results seem to be significantly different (Table 2: Binomial Distribution). Discussion on why this happened (whether if it was due to the randomness of the seed as stated in the discussion) would have been good.

Questions
- What was the motivation behind implementing the code from C++ to Python, especially since the C++ compiler is more efficient?
- Author seems to use . for commas, when discussing numbers (ex line 170); commas are more commonly used instead though.

---

### Meta-Review · Area_Chair_w5bU · 2022-04-08

**Recommendation:** Accept
**Confidence:** 4

**Metareview:**

A great study and the author manages to validate the claims made in the original paper. It would be helpful if the authors have any insights on the results which are different from the original paper.

---

### Decision · Program_Chairs · 2022-04-09

**Decision:**

Accept

**Comment:**

Following the recommendation of reviewers and meta-reviewer, the paper is accepted for ML Reproducibility Challenge 2021, and will be published in the upcoming special edition of ReScience Journal.